# Platelet Membrane Proteins as Pain Biomarkers in Patients with Severe Dementia

**DOI:** 10.3390/biomedicines11020380

**Published:** 2023-01-27

**Authors:** Hugo Ribeiro, Raquel Alves, Joana Jorge, Ana Cristina Gonçalves, Ana Bela Sarmento-Ribeiro, Manuel Teixeira-Veríssimo, Marília Dourado, José Paulo Andrade

**Affiliations:** 1Community Support Team in Palliative Care—Group of Health Centers Gaia, 4430-999 Vila Nova de Gaia, Portugal; 2Faculty of Medicine, University of Porto, 4200-319 Porto, Portugal; 3Faculty of Medicine, University of Coimbra, 3000-548 Coimbra, Portugal; 4 Coimbra Institute for Clinical and Biomedical Research (iCBR)—Group of Environment, Genetics and Oncobiology (CIMAGO), University of Coimbra (FMUC), 3000-548 Coimbra, Portugal; 5 University Clinics of Hematology and Oncology and Laboratory of Oncobiology and Hematology (LOH), Faculty of Medicine, University of Coimbra (FMUC), 3000-548 Coimbra, Portugal; 6Center for Innovative Biomedicine and Biotechnology (CIBB), 3000-548 Coimbra, Portugal; 7Hematology Service, Centro Hospitalar e Universitário de Coimbra (CHUC), 3000-548 Coimbra, Portugal; 8CINTESIS@RISE, Faculty of Medicine, University of Porto, 4200-319 Porto, Portugal; 9Unit of Anatomy, Department of Biomedicine, Faculty of Medicine, University of Porto, 4200-319 Porto, Portugal

**Keywords:** chronic pain, pharmacology, platelets, biomarkers, palliative care

## Abstract

Pain is one of the most frequent health problems, and its evaluation and therapeutic approach largely depend on patient self-report. When it is not possible to obtain a self-report, the therapeutic decision becomes more difficult and limited. This study aims to evaluate whether some membrane platelet proteins could be of value in pain characterization. To achieve this goal, we used 53 blood samples obtained from palliative patients, 44 with non-oncological pain and nine without pain. We observed in patients with pain a decrease in the percentage of platelets expressing CD36, CD49f, and CD61 and in the expression levels of CD49f and CD61 when compared with patients without pain. Besides that, an increase in the percentage of platelets expressing CD62p was observed in patients with pain. These results suggest that the levels of these platelet cluster differentiations (CDs) could have some value as pain biomarkers objectively since they are not dependent on the patient’s participation. Likewise, CD40 seems to have some importance as a biomarker of moderate and/or severe pain. The identification of pain biomarkers such as CD40, CD49f, CD62p and CD61 can lead to an adjustment of the therapeutic strategy, contributing to a faster and more adequate control of pain and reduction in patient-associated suffering.

## 1. Introduction

The International Association for the Study of Pain (IASP) defines pain as an “unpleasant sensory and emotional experience associated with, or resembling that associated with, actual or potential tissue damage” [1]. Pain is multifactorial, with multiple pathways involved [2], which explains why it is a multidimensional experience that can significantly impair an individual’s quality of life [3]. Pain can be classified according to tissue damage (nociceptive pain), nerve damage (neuropathic pain), and altered pain modulation (nociplastic pain) [3].

Pain characterization is essential for the correct approach for patients with chronic pain, starting with the assessment of pain intensity [4]. The proper characterization is dependent on patient self-report; pain multidimensional hetero-assessment scales can be used as an alternative, especially for dementia patients with and/or patients that cannot characterize their pain [5,6,7]. However, as pain is a sensory and emotional experience, it is subjective, and its expression is primarily determined by the perceived intensity of the painful sensation [5].

Preclinical and clinical studies have investigated the hypothesis that biomarkers may also be used to identify and quantify pain. Findings from a preclinical study show that inflammatory pain and neuropathic pain have different biomarkers [8], but most studies do not correlate with pain duration or intensity. Further studies are needed to gain insights into pain biomarkers to enhance pain management practices improving patient care, especially for those who suffer from severe cognitive decline or dementia and are unable to express themselves.

Platelet heterogeneity and subpopulations may suggest distinct biological roles for different platelet subpopulations and may be useful in evaluating inherited or acquired platelet disorders and platelet function in health and disease [9]. Besides their role in hemostasis, thrombosis, and wound healing, platelets are now known to play major effector activities in several additional functions, including inflammatory reactions and innate immune responses [9]. Further, platelets are the closest and most accessible peripheral neuronal-like cellular system likely to provide a wealth of information about neuronal functioning [10]. 

Since discovered by Giulio Bizzozero in 1882 [11], platelets have been exploited for their clinical value. Platelet surface receptors, such as CD62p (P-selectin) and CD41 (GPIIb-IIIa), have also been quantified as markers of the activation state of platelets [12]. The extravascular activation of platelets may contribute to nociceptor excitation and pain since platelets store and, upon stimulation, release potential allogenic substances such as serotonin, histamine, and precursor molecules of bradykinin [13,14].

Identifying blood and platelet pain biomarkers has been advanced as the next great tool for pain identification and characterization, allowing tailored treatments [15,16].

Few studies relate pain with platelet activation, as pain inhibition occurs with platelet antiaggregants [17,18,19,20]. However, there are other studies on specific situations, for instance, knee osteoarthritis or post-teeth extraction pain, where platelet-rich plasma (PRP) or fibrin-rich platelets (PRF) may have a significant analgesic effect [21,22,23,24]. However, the role of peripheral blood platelets membrane proteins as markers for pain evaluation and characterization is not yet clarified [25,26].

Here, we evaluated the levels of membrane proteins, namely receptors related to several recognized functions of platelets, such as recognition, adhesion, aggregation, activation, inflammation, and immune modulation. In this context, the levels of the glycoprotein IV (CD36), the adhesion molecules, integrin α6 (CD49f), integrin β3 (CD61) and p-selectin (CD62p), the complement activation inhibitor protein (CD59) and the TNFα family receptor CD40 were evaluated in palliative patients platelets.

The main aim of this study is to evaluate whether some known platelet membrane proteins could be of value in pain characterization and can be used as non-invasive pain biomarkers in patients where we cannot rely on their self-report, particularly patients with advanced dementia.

## 2. Materials and Methods

For this study, we collected individual and clinical data and peripheral blood samples from 53 palliative patients with non-oncological diseases, followed by a specialized palliative care team between 1 September and 31 December 2021. 

This is an observational, analytic, transversal, non-interventional study using medical and nursing records on chronic pain patients.

The Ethics Committees of the Faculty of Medicine of the University of Porto and the North Regional Health Administration of Portugal approved the research procedures, and the study was conducted following the Declaration of Helsinki. Before enrollment, participants or their legal representatives provided informed consent for participation. The international ethical guidelines of confidentiality, the anonymity of personal data, and the abandonment option were followed.

For the collection of individual and clinical data, we consulted the records in the individual clinical files, after which they were registered in a protected Microsoft Excel sheet. To identify each patient, an alphanumeric code was used, thus keeping the identity confidential since only the researcher knows it. Following the European General Data Protection Regulation (GDPR), these electronic files will be deleted after the end of the study and the publication of the results. Data were collected regarding the following variables: age, sex, type and intensity of pain, opioid and other analgesics, such as nonsteroidal anti-inflammatory drugs (NSAIDs) and acetaminophen, and doses in use. We also noted whether the patient had pain control at the moment of blood sample collection.

We also collected information regarding the presence of a diagnosis of dementia and its type. For patients that had severe dementia, we used the Pain Assessment in Advanced Dementia Scale (PAINAD) [27] to identify and distinguish those who probably have uncontrolled pain, and we separated PAINAD values (under 5, between 5 and 7 and 8 to 10), to evaluate a possible clinical correlation of these PAINAD values and mild, moderate or severe pain, respectively. For patients who could self-report their pain, we used the pain numeric scale [28].

For the purpose of analyzing the expression of the platelet biomarkers, we selected the following: glycoprotein IV (CD36), integrin α6 (CD49f), integrin β3 (CD61), and p-selectin (CD62p), the complement activation inhibitor protein (CD59) and TNFα family receptor CD40, that we analyzed in absolute and relative concentration.

We proceeded to the ROC curve analysis to evaluate the significance of the area under the curve and if we have a cut-off point for CD40.

### 2.1. Platelets Membrane Proteins Evaluation by Flow Cytometry

The platelet phenotyping was performed in freshly prepared platelet suspensions. None of the participants were recently transfused. Briefly, platelets were separated by centrifuging citrated blood specimens at 120× *g* for 20 min at room temperature (RT). Then, the platelet-rich plasma was collected into a tube, and the platelets were washed using 2 mL of wash buffer (BD cell wash). Next, 1 × 10^6^ platelets were incubated with the monoclonal antibodies anti-CD36 conjugated with FITC (BD Pharmingen, BD Biosystems, San Diego, CA, USA), anti-CD49f conjugated with PE conjugate (BD Pharmingen, BD Biosystems), anti-CD61 conjugated with PerCP-Cy5.5 (BD Pharmingen, BD Biosystems), anti-CD62 conjugated with APC (BD Pharmingen, BD Biosystems), anti-CD59 conjugated with BV421(BD Horizon, BD Biosystems), and anti-CD40 conjugated with BV510 (BD Horizon, BD Biosystems) for 15 min at RT in the dark, according to manufacture instructions. Then, cells were washed twice with FACS flow (BD Biosystems) by centrifugation at 300× *g* for 5 min and immediately analyzed in a FACS Canto II flow cytometer (BD Biosystems). At least 50,000 events were collected using FACS DIVA software (BD Biosystems), and the results were analyzed through Infinicyte software (Cytognos, Salamanca, Spain). The results are expressed in the percentage of cells expressing each protein marker and as mean fluorescence intensity (MIF). 

### 2.2. Statistical Analysis

Statistical analysis was performed using the SPSS software (Statistical Package for the Social Sciences, version 28.0 for Windows, IBM Corp., Armonk, NY, USA). For the statistical analysis of data, we used measures of descriptive statistics (absolute and relative frequencies, means, and respective standard deviations) and inferential statistics. The significance level for rejecting the null hypothesis was set at (α) ≤ 0.05. We used the Pearson correlation coefficient, Fisher’s test, Chi-squared test of independence, Student’s *t*-test for independent samples, Mann–Whitney U test, and Kruskal–Wallis test. The normality of distribution was analyzed with the Shapiro-Wilk test, and the homogeneity of variances was analyzed with Levene’s test. We analyzed the Chi-squared assumption that there should not be more than 20% of cells with expected frequencies inferior to 5. In those situations where this assumption could not be satisfied, we used the Chi-squared test with the Monte Carlo simulation.

The following variables were included: age, sex, type of pain, the intensity of pain, opioid and dose used, and other analgesics, such as nonsteroidal anti-inflammatory drugs (NSAIDs) and paracetamol, and absolute (MIF) and relative concentration levels of CD36, CD49f, CD61, CD62p, CD59, and CD40. We also included the type of dementia (because most of the patients had this condition) and controlled pain time for patients diagnosed with chronic pain, but their pain is controlled at the moment of blood collection. For patients that had severe dementia, we used the Pain Assessment in Advanced Dementia Scale (PAINAD) [27] to identify uncontrolled pain. Other variables that could have a relationship with our findings were also included, such as renal function (using the Cockcroft–Gault formula [29], body mass index (BMI), functionality (using the Karnovsky scale [30]), and nutritional status (using Mini-Nutritional Assessment scale [31]).

### 2.3. Inclusion Criteria

All patients under clinical follow-up from a palliative care specialized team from the North region of Portugal with non-oncological diseases. 

All patients or their legal representatives provided informed consent for participation.

## 3. Results

We selected 95 patients. However, five patients or their legal representatives refused to participate in this study and 20 of them were in their last days of life, and it was decided not to conduct a blood collection in this clinical condition. We could not collect blood samples from 17 patients, as they were too fragile, with hypovolemia and bad venous accesses. 

So, we collected samples from a total of 53 patients with an average age of 74.8 years old, a minimum of 29 and a maximum of 98 years old; most were female [*n* = 39 (73.6%)]. Forty-four patients suffered from pain, and nine had no pain. We had no patients with known autoimmune diseases (inflammatory bowel disease, multiple sclerosis, psoriasis, or other of these conditions).

Among the 44 patients with chronic pain, 38 had severe dementia. We could not evaluate the type of pain and intensity in these patients. Therefore, we used PAINAD to distinguish those who probably have uncontrolled pain (PAINAD ≥ 5, present in 32 patients), and we separated PAINAD values (under 5, between 5 and 7 and 8 to 10) because in the clinical evaluation, there was a correlation between these PAINAD values and the possibility of having mild, moderate or severe pain, respectively. Among the 44 patients with pain, 19 were under opioid treatment (15 with dementia and four without dementia).

We had six patients with pain and without dementia and nine without pain and without dementia (controls), as shown in Table 1.

In the platelets of patients with chronic pain, we observe a statistically significant decrease in the percentage of platelets expressing CD36, CD49f, and CD61, but the decrease in the expression levels of these biomarkers was only observed for CD49f and CD61, compared with those patients without pain (Table 2). An increase in the percentage of platelets expressing CD62p was detected when compared with patients without pain (*p* = 0.002) (Table 2). 

While the differences observed in CD36, CD49f and CD61 are age and sex independent, the difference observed in CD62p were more accentuated in men (about three times higher) comparatively with the observed in women (Table 3). When we compared the platelets phenotype between patients with dementia with those without dementia, we did not observe any statistically significant difference. Further, the relationship between the type of pain and the type of dementia is also not statistically significant (Table 3).

As we can observe in Table 4, patients under opioids, when compared with patients that did not receive opioids, present a statistically significant decrease in the percentage of platelets expressing CD36 (88.52% vs. 95.58, *p* = 0.026) and in the expression levels of CD49f (4254.22 vs. 4995.90, *p* = 0.012), CD61 (12643.79 vs. 15128.72, *p* = 0.029) and CD59 (1345.35 vs. 1975.92, *p* = 0.008). 

Regarding the type of pain and its relationship with biomarkers under study, we found statistically significant differences, as shown in Table 5. A decrease in the expression levels of CD62p in the platelets of patients with nociceptive pain was observed compared with patients presenting mixed pain (723.52 vs. 1953.63, *p* = 0.002) (Table 5).

The clinical history (other diseases) of the patients does not seem to interfere with platelet biomarkers.

With regard to pain intensity, we observed a significant relationship with the percentage of platelets expressing CD40. In fact, the percentage of platelets expressing CD40 is significantly higher in patients with moderate-severe pain (considering PAINAD ≥5) compared with those with mild pain (0.18% vs. 0.02%, *p* = 0.047), as it is shown in Table 6. 

We used receiver operating characteristic curve (ROC) analysis to verify if CD40 could be a peripheral biomarker of pain intensity. We proceed to the ROC curve analysis to evaluate the significance of the area under the curve and if we have a cut-off point for this biomarker. The area under the curve is statistically significant, 0.777, with *p* = 0.049, as shown in Figure 1 and Table 7.

The ideal cut-off point corresponds to 0.025 (sensitivity 74.2%, specificity 80%, positive predictive value = 95.8%, negative predictive value = 33.3%).

## 4. Discussion

In patients with severe dementia, we cannot rely on self-report, and the characterization of the pain is difficult. The identification of biomarkers could be a valuable solution to enable targeted medical treatment. In this study, we find that CD36, CD49f, percentage of CD49f, CD61, percentage of CD61, and CD62p can be considered as platelet biomarkers of pain, CD62p as a platelet biomarker for nociceptive pain and CD40 as a platelet biomarker for moderate-severe pain.

The hypothesis that biomarkers may be used to identify and quantify pain was investigated in several preclinical and clinical studies. A preclinical study showed that inflammatory and neuropathic pain have different biomarkers [8]. Further investigations provided mixed results. For example, cystatin C levels in the cerebrospinal fluid appear to be a predictive marker for postherpetic neuralgia in patients with varicella-zoster virus and a pain marker in women experiencing labor pain. However, it is not correlated with pain duration or intensity. Investigations into potential biomarkers for chest pain showed that cardiac markers used to aid in the diagnosis and prognosis of cardiac disease correlate with tissue damage rather than with pain [8]. Further studies are needed to gain insights into biomarkers for pain to enhance pain management practices.

Platelet receptors are important for their normal functioning as they either activate platelets or act as adhesion molecules interacting with the damaged endothelium, other platelets, and leukocytes. Besides the platelet role in hemostasis, they also have a role in inflammation, antimicrobial activity, angiogenesis, tumor growth, and metastasis. In the absence of their receptors, platelets are unable to perform these functions. Some of the well-recognized platelet receptors are integrins, leucine-rich repeats receptors, selectins, tetraspanins, transmembrane receptors, prostaglandin receptors, lipid receptors, immunoglobulin superfamily receptors, tyrosine kinase receptors, and other platelet receptors [32]. The present study focused on membrane glycoproteins, such as CD36, CD49f, CD61, CD62p, CD59, and CD40.

CD36 or Glycoprotein IV is a member of the class B scavenger receptor family of cell surface proteins, a multiligand pattern recognition receptor that interacts with a large number of structurally dissimilar ligands, including long chain fatty acid (LCFA), advanced glycation end products (AGE), thrombospondin-1, oxidized low-density lipoproteins (oxLDLs), high density lipoprotein (HDL), phosphatidylserine, apoptotic cells, beta-amyloid fibrils (fAβ), collagens I and IV, and Plasmodium falciparum-infected erythrocytes [33]. 

CD49f is an adhesion molecule, namely α6-integrin 1, associated with inflammation towards the regulation of differentiation, adhesion, and migration of human mesenchymal stem cells [34]. CD61 is the integrin β3, a glycoprotein that plays a role in platelet aggregation and also as a receptor for fibrinogen, fibronectin, von Willebrand factor and vitronectrin [35]. CD62p or P-selectin is a membrane protein that redistributes to the plasma membrane during platelet activation and degranulation and mediates the interaction of activated endothelial cells or platelets with leukocytes [36]. CD59 is the complement activation inhibitor protein that binds to complement components C5 and C9 and prevents the polymerization of C9, which is required for the formation of the membrane attack complex (MAC) [37]. CD40 is a receptor of the TNFα family, being a cytokine produced by many cells, and was originally identified by its cytotoxic effects. In addition to inducing cell death in some types of cells, it also elicits a wide range of physiological responses, such as inflammation, cell proliferation, and differentiation [38]. Further, circulating biomarkers of platelet activation, including soluble CD40 and CD62p, have also been studied as a strategy to monitor the efficacy of combination antiretroviral therapy (cART) in patients infected with HIV [39]. 

Platelet surface receptors have also been quantified as markers of the activation state of platelets, and platelet activation is increased in dementia [40]. Both platelet CD62p (P-selectin) expression and CD41 (GPIIb-IIIa) complex activation are significantly elevated in Alzheimer’s Dementia (AD) patients [41]. CD41 is a platelet activation marker in dementia, and the increase in CD41 complex expression in platelets was associated with faster cognitive decline in AD [41]. However, there is an overwhelming lack of additional clinical studies. Another platelet receptor, CD62p, is present on activated platelet membranes [42] and promotes platelet adhesion and thrombin formation [43]. Increased levels of CD62p were found in circulation in AD patients, but there was no significant change in the membrane-bound [12,38]. The soluble form of CD62p was also significantly elevated in HIV patients not under cART, compared with those cART-treated and with healthy control groups, suggesting its role in monitoring combination antiretroviral therapy [39]. 

We already know that there is platelet activation in pain, especially inflammatory pain (Barkai et al., 2019; Beurling-Harbury and Schade, 1989), but we are still unaware of all the pathophysiology and molecular biology involved [19]. The identification of CD36, CD49f, percentage of CD49f, CD61, percentage of CD61, and CD62p as platelet biomarkers of pain was statistically significant. These new markers must be considered in the process of obtaining pain biomarkers in peripheral blood.

For patients with well-controlled pain, in comparison with those with non-controlled pain, we did not find statistically significant differences regarding laboratory data, including patients with well-controlled pain for seven or more days. These data are relevant because we could expect a reduction in platelet activation with pain control, considering that the half-life of platelets is seven days. However, it may mean that some underlying mechanisms and causes of pain may be equally active, and there is only pain desensitization. 

In opioid-treated patients presenting a better control of pain, there is less expression of biomarkers such as CD36, CD49f, CD61, and CD59. These data are in agreement with the literature on the effect of opioids on platelets, particularly in reducing their activity [44,45,46].

The sample was too small to assess more types of pain in addition to nociceptive and mixed pain. However, CD62p has been shown to be a reliable marker of nociceptive pain. Although it has not yet been specifically studied in this regard, the role of p-selectin is already known and has been linked to inflammatory pain [47,48].

CD40 (TNFα receptor family) was identified as a potential biomarker of moderate or severe pain intensity. The association of a biomarker with pain severity has not yet been established, although there are several references to platelet markers and disease severity [49,50,51]. In addition to the biomarkers of the existence or not of pain, which can be a powerful aid in the therapeutic approach of patients, the identification of biomarkers that allow the characterization of pain, in this case by intensity, can allow the therapeutic adjustment as quickly as possible, reducing suffering and minimizing the adverse effects of drugs.

Further, and besides the involvement of platelets in several diseases the presented markers, such as CD40, in autoimmune diseases such as inflammatory bowel disease, multiple sclerosis, psoriasis, or other autoimmune diseases, we had no patients with these diseases. Other diseases from the known medical history of these patients (such as hypertension) did not influence the platelet markers, according to what we found in the literature [9,10,12,17,26].

This is one of the largest studies on pain biomarkers that we know of, and it is the only study comparing patients with non-oncological pain with specific platelet biomarkers. However, the sample is small, and further studies are needed considering these markers to confirm their viability as pain markers and CD40 as a marker of moderate-severe pain. 

In vulnerable and dependent patients, in whom we cannot rely on self-report, the identification of pain biomarkers such as CD40 can lead to an adjustment of the therapeutic strategy, according to the WHO ladder [4], contributing to a faster and more adequate control of pain and reduction in associated suffering.

## Figures and Tables

**Figure 1 biomedicines-11-00380-f001:**
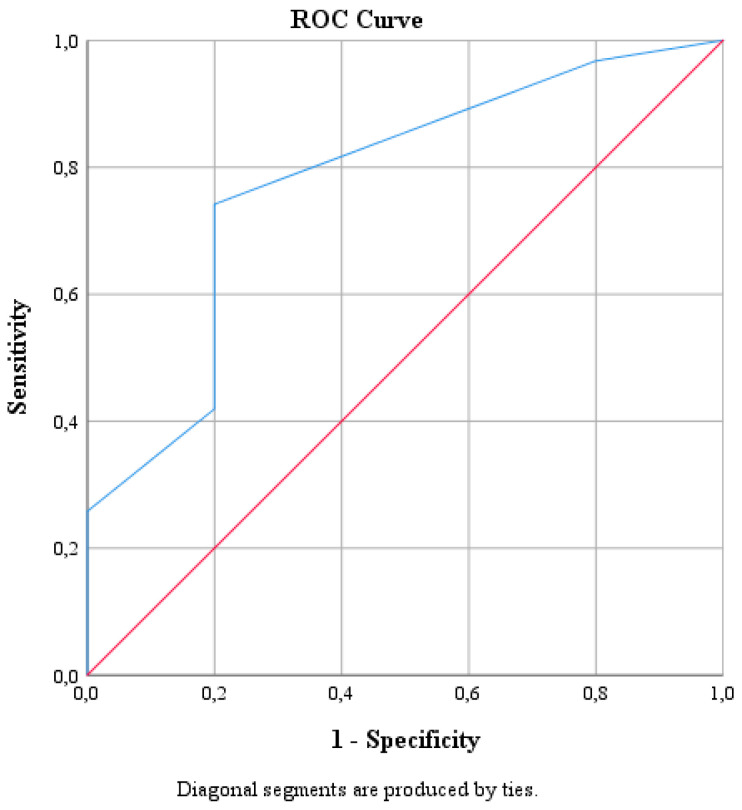
ROC curve of CD 40 as a biomarker of moderate or severe pain.

**Table 1 biomedicines-11-00380-t001:** Characterization of the Study Population.

Total of Patients: 53		Patients with Pain and Dementia (*n* = 38)	Patients with Pain and without Dementia (*n* = 6)	Patients without Pain and without Dementia (Controls) (*n* = 9)
**Gender**	**Male**	29% (*n* = 11)	50% (*n* = 3)	0
	**Female**	71% (*n* = 27)	50% (*n* = 3)	100%
**Average age (years)**		84.1	62.5	44.7
**PAINAD**	**<5**	15.8% (*n* = 6)	NA	NA *
	**5–7**	73.7% (*n* = 28)	NA	NA
	**8–10**	10.5% (*n* = 4)	NA	NA
**Average numeric pain scale**		NA	4.3	0
**Type of Pain**	**Nociceptive**	39.5% (*n* = 15)	50% (*n* = 3)	NA
	**Neuropathic**	2.6% (*n* = 1)	16.7% (*n* = 1)	NA
	**Mixed**	57.9% (*n* = 22)	33.3% (*n* = 2)	NA
**Type of Dementia**	**Vascular**	36.8% (*n* = 14)	NA	NA
	**Alzheimer**	36.8% (*n* = 14)	NA	NA
	**Mixed**	7.9% (*n* = 3)	NA	NA
	**Other**	18.4% (*n* = 7)	NA	NA
**Under opioid treatment**		39.5% (*n* = 15)	66.7% (*n* = 4)	NA

* NA: Not Applicable.

**Table 2 biomedicines-11-00380-t002:** Platelets Membrane Proteins in Patients with and without Chronic Pain.

	Patients without Pain and without Dementia Controls (*n* = 9)	Patients with Pain (*n* = 44)	
	M ± SD	M ± SD	*p*
CD36 (%)	98.04 ±1.97	91.54 ± 12.30	0.004 **
CD36 (MIF)	6475.64 ± 2081.58	5506.31 ± 2168.98	0.228
CD49f (%)	99.16 ± 0.35	95.31 ± 10.79	0.037 *
CD49f (MIF)	5385.83 ± 855.09	4540.24 ± 969.50	0.021 *
CD61 (%)	99.37 ± 0.18	95.28 ± 10.76	0.026 *
CD61 (MIF)	16561.81 ± 1404.83	13571.25 ± 4064.78	0.009 **
CD62P (%)	8.47 ± 4.87	22.62 ± 23.65	0.002 **
CD62P (MIF)	760.75 ± 114.10	1357.76 ± 2606.10	0.499
CD59 (%)	4.76 ± 3.03	3.45 ± 5.67	0.507
MIF CD59 (MIF)	1556.62 ± 509.90	1771.15 ± 858.56	0.478
CD40 (%)	0.15 ± 0.09	0.09 ± 0.17	0.289
CD40 (MIF)	1476.08 ± 352.14	1456.12 ± 501.07	0.911

MIF—Mean Fluoresce intensity; * *p* ≤ 0.05, ** *p* ≤ 0.01. F: median intensity fluorescence; M: mean; SD: standard deviation; *p*: significance.

**Table 3 biomedicines-11-00380-t003:** Gender And Membrane Protein Platelets.

	Female (*n* = 39)	Male (*n* = 14)	
	M ± SD	M ± SD	*p*
% CD36	92.89 ± 10.89	92.58 ± 13.25	0.899
MIF CD36	5392.26 ± 2050.79	6662.28 ± 2330.91	0.090
% CD49f	96.35 ± 8.80	95.18 ± 12.86	0.445
MIF CD49f	4617.57 ± 978.43	4986.02 ± 1057.26	0.291
% CD61	96.38 ± 8.80	95.13 ± 12.83	0.272
MIF CD61	14465.44 ± 3481.59	13172.91 ± 4992.24	0.525
**% CD62P**	**13.76 ± 17.43**	**39.24 ± 24.50**	**0.006 *****
MIF CD62P	829.68 ± 261.73	2549.57 ± 4691.50	0.251
% CD59	3.74 ± 5.35	3.59 ± 5.20	0.839
MIF CD59	1643.84 ± 790.28	2000.70 ± 815.41	0.130
% CD40	0.09 ± 0.10	0.14 ± 0.28	0.939
MIF CD40	1471.59 ± 505.76	1423.23 ± 362.89	0.899

*** *p* ≤ 0.001; MIF: median intensity fluorescence; M: mean; SD: standard deviation; *p*: significance; bold: values statistically significant.

**Table 4 biomedicines-11-00380-t004:** Comparative Analysis Of Membrane Platelets Proteins Between Patients Receiving And Not Receiving Opioids Medication.

	No Opioids (*n* = 25)	Opioids (*n* = 19)	
	M ± SD	M ± SD	*p*
**% pos CD36**	**95.58** **± 8.38**	**88.52** **± 14.02**	**0.026 ***
MIF CD36	6007.26 ± 2091.56	5211.73 ± 2246.28	0.228
% pos CD49f	98.82 ± 2.39	91.78 ± 14.55	0.070
**MIF CD49f**	**4995.90** **± 850.84**	**4254.22** **± 1065.23**	**0.012 ***
% pos CD61	98.87 ± 2.40	91.73 ± 14.51	0.077
**MIF CD61**	**15128.72** **± 3014.28**	**12643.79** **± 4616.87**	**0.029 ***
% pos CD62P	21.08 ± 22.70	17.93 ± 21.35	0.290
MIF CD62P	1483.69 ± 2983.98	863.37 ± 388.08	0.242
% pos CD59	2.94 ± 3.33	4.90 ± 7.28	0.597
**MIF CD59**	**1975.92** **± 886.94**	**1345.35** **± 444.10**	**0.008 ****
% pos CD40	0.07 ± 0.08	0.15 ± 0.23	0.742
MIF CD40	1485.36 ± 467.55	1420.60 ± 490.21	0.458

* *p* ≤ 0.05, ** *p* ≤ 0.01; MIF: median intensity fluorescence; M: mean; SD: standard deviation; *p*: significance; bold: values statistically significant.

**Table 5 biomedicines-11-00380-t005:** Membrane Proteins Platelets In Patients Suffering From Nociceptive Pain Or Mixed Pain.

	Nociceptive (*n* = 18)	Mixed (*n* = 26)	
	M ± SD	M ± SD	*p*
% pos CD36	91.17 ± 11.90	92.52 ± 10.84	0.762
MIF CD36	4751.72 ± 1640.23	5952.76 ± 2468.46	0.118
% pos CD49f	94.47 ± 12.16	97.46 ± 5.19	0.789
MIF CD49f	4206.69 ± 995.35	4859.81 ± 890.59	0.056
% pos CD61	94.37 ± 12.13	97.45 ± 5.15	0.762
MIF CD61	12507.53 ± 5215.25	14025.16 ± 3190.66	0.682
% pos CD62P	13.89 ± 14.69	31.73 ± 28.35	0.117
**MIF CD62P**	**723.55 ± 184.49**	**1953.63 ± 3679.90**	**0.002 ****
% pos CD59	4.49 ± 6.59	3.05 ± 5.46	0.789
MIF CD59	1570.69 ± 826.36	1963.51 ± 937.17	0.274
% pos CD40	0.16 ± 0.25	0.04 ± 0.02	0.145
MIF CD40	1530.59 ± 601.99	1466.89 ± 448.98	0.986

** *p* ≤ 0.01, MIF: median intensity fluorescence; M: mean; SD: standard deviation; *p*: significance; bold: values statistically significant.

**Table 6 biomedicines-11-00380-t006:** Membrane Proteins Platelets In Patients Suffering From Mild Pain And With Moderate Or Severe Pain.

	Mild Pain (PAINAD < 5)	Moderate or Severe Pain (PAINAD 5-10)	
	M	M	*p*
% pos CD36	90.34 ± 18.24	92.88 ± 9.64	0.563
MIF CD36	5635.97 ± 2538.02	5424.41 ± 2158.15	0.859
% pos CD49f	99.19 ± 0.66	95.94 ± 9.29	0.625
MIF CD49f	4310.17 ± 1200.38	4549.61 ± 949.86	0.723
% pos CD61	99.16 ± 0.66	95.90 ± 9.27	0.625
MIF CD61	10900.50 ± 4611.48	13915.31 ± 3929.31	0.059
% pos CD62P	16.62 ± 21.39	23.77 ± 24.53	0.690
MIF CD62P	3966.84 ± 7066.90	918.62 ± 383.06	0.533
% pos CD59	1.25 ± 1.49	3.90 ± 6.08	0.282
MIF CD59	1699.58 ± 1043.69	1793.88 ± 856.69	0.533
**% pos CD40**	**0.02 ± 0.02**	**0.10 ± 0.18**	**0.047 ***
MIF CD40	1202.00 ± 232.79	1509.31 ± 524.97	0.207

* *p* ≤ 0.05,MIF: median intensity fluorescence; M: mean; SD: standard deviation; *p*: significance; bold: values statistically significant.

**Table 7 biomedicines-11-00380-t007:** ROC Analysis Data for CD40.

Area	SE ^a^	Asymptotic *p* ^b^	Asymptotic 95% Confidence Interval	
			Lower Bound	Upper Bound
0.777	0.114	0.049	0.555	1.000

^a^: Standard Error; ^b^: Significance.

## Data Availability

The data presented in this study are available on request from the corresponding author.

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
