# Peer review of "Platelet Membrane Proteins as Pain Biomarkers in Patients with Severe Dementia"

_biomedicines, 2023, doi:10.3390/biomedicines11020380_

Round 1

Reviewer 1 Report

1. Please provide the inclusion and exclusion criteria. In my opinion, the presented by the Authors are not enough and should also involved the presence of variosu type of diseases and or disorders for which some of the presented markers may play a crucial role (e.g., CD40 is knwow for its role in autoimmune diseases as well as diabetes, etc.). Indeed, conditions other than pain may have similar results of platelet studies.

2. I assume that the group patients (n=9) without pain was control? However, this group includes only females with an average year of 44.7 in contrast to mostly male-reached group with an average year of 62-84. Unfortunately, in my opinion, such a comparison to the control group does not provide reliable results.

3. I'm not sure about the table 2. The Authors presented results for the entire participants, also healthy ones, which were mixed with other groups of patients suffering from pain. What was the reason to do that? Such results do not provide any information. The control gruop shoould be excluded from these results. 

4. Some punctuation errors can be found.

Reviewer 2 Report

Lines 164-166:  This study has o very small number of participants and  the group with pain versus the group without pain is disproportionate 44 vs 9.

CD40L in particular, as well as sCD62P are known as pro-inflammatory factors. (Steel HC, Venter WDF, Theron AJ, Anderson R, Feldman C, Arulappan N, Rossouw TM. Differential Responsiveness of the Platelet Biomarkers, Systemic CD40 Ligand, CD62P, and Platelet-Derived Growth Factor-BB, to Virally-Suppressive Antiretroviral Therapy. Front Immunol. 2021 Jan 29;11:594110. doi: 10.3389/fimmu.2020.594110. PMID: 33584658; PMCID: PMC7878378.)

Lines 88-89: In this study include 53 palliative patients, the pathologies of these patients and what may be the cause of the pain are not specified.  

This study must be completed with a much larger number of patients, in whom the presence of pain must be well determined, more data related to pain, with patients who can correctly diagnose the presence of pain.

Otherwise, these correlations cannot be made specifically on pain but on the inflammatory process.

Round 2

Reviewer 1 Report

The Authors provided sufficient responses to the questions and suggestions made by the Reviewer.

Author Response

Dear reviewer,

Thank you very much for reviewing our manuscript. Your insights led to improvement of our paper. We will continue our research.

Reviewer 2 Report

I appreciate the work of the authors for this research, but to make new statements related to the involvement of platelet membrane proteins as pain biomarkers, a lot of data is needed. You will probably continue researching this hypothesis. Good luck in further research!

Author Response

(The authors gave the same response as above.)
